# Exploring the role of human resource development functions on crisis management: The case of Dubai-UAE during Covid-19 crisis

**Amir Hamad Salim Binnashira Alketbi**[1◉], **Juan Antonio Jimber del Rio**[1◉]*,
**Alberto Ibáñez Fernández**[2◉]

1 Department Agricultural Economics, Finance, and Accounting, University of Cordoba, Córdoba, Spain,
2 Management, International Relations, University of Science and Technology, Fujairah, United Arab Emirates

◉ These authors contributed equally to this work.
* jjimber@uco.es

**Data Availability Statement:** All relevant data are within the manuscript and its Supporting Information files.

## Abstract

Employee welfare represents a critical element of success for companies to remain competitive. Human resources increasingly encompass the management of critical situations that affect the employees' wellbeing. This research analyzes the effect of Human Resource Development (HRD), functions on the effectiveness of crisis management. It is an attempt to include HRD in the theory of Crisis management. Using Structural Equation Models-Partial Least Squares (SEM-PLS) analysis, the study analyzes how training, leadership, organizational strategy, and organizational culture directly positively impact the efficiency of Crisis management (CM) during the Covid-19 crisis in the public entities of Dubai-UAE. In particular, training showed to be the best predictor, followed by the Organizational culture. Organizational structure, Values and uniqueness show no impact on CM within the context of public entities of Dubai-UAE.

## Introduction

Human Resource Development (HRD) is defined as "the organized learning venture displayed within an organization to enhance personal growth and professional performance of the employees". HRD aims to improve organizational efficiency to enhance the individual, the job and the organization [1]. In the contemporary corporate world, the role of Human Resources (HR) has changed to accommodate different needs and developments within the workplace [2]. Mishra [3] stated that the organizations that involve employee welfare as a crucial aspect of crisis management are successful in the long run.

Involvement in crisis management and efficient and active responsibility in planning and training for any crisis management can be considered significant HR development. In this current crisis, due to the Covid-19 pandemic, both employers and employees are exposed to

**Funding:** The author(s) received no specific funding for this work.

**Competing interests:** The authors have declared that no competing interests exist.

extreme risks of financial insecurity in alignment with a mental health concern, loss of well-being and emotional stress. In the UAE, more than 23% of the employees have reported unmanageable work stress as working from home is challenging in a fragile business condition during this crisis period [4]. The Ministry of Human Resources and Emiratisation (MoHRE) have introduced two solutions during the Covid-19 crisis, namely Redundancy and Restructuring and Remote working Policy [5]. Such policies aim to provide additional confidence and guidance concerning accurate HR measures that can efficiently implement across the UAE workforces. This requires the HRD to play pivotal roles in protecting employees' welfare from pre-crisis to post-crisis phases, thereby facilitating organizational sustainability. During a crisis period, the HRD of the UAE corporate firms must ensure that the workforce is prepared for the crisis through adequate training, effective empowerment, and leadership approaches, assuring safety and security initiatives and effective communication strategies in promoting crisis management [4].

The significance of HRD came into the limelight when businesses started to expand beyond the domestic borders due to globalization. This highlighted the competency of HRD in marinating a robust and efficient workforce to support the needs and requirements of the organization and respond successfully to the mission and vision for the long term [6]. The emphasis sizes the workforce's training and development, organizational development by implementing micro or macro-level changes and supporting career planning for the staff. Private and public-sector employers have applied strategic measures to manage employee turnover during a crisis period. This includes proper training to fight the crisis scenario, grooming of the talents, implementing organizational restructuring to benefit both the workforce and administration and others. An extensive literature supports the statements described above concerning the importance of HRD in crisis management [7]. However, the Covid-19 pandemic has created a financial and emotional crisis. The world has witnessed significant closure of established ventures leading to the loss of millions of jobs and increased unemployment. Thus, shifting from a global pandemic crisis scenario and narrowing down the research to crisis management in the UAE by implementing strategic HRD measures, the article highlights the HR policies to support the UAE workforce during the Covid-19 crisis [8].

The Covid-19 crisis is a recent incidence; however, businesses worldwide have faced different types of crises, which have disrupted their business model. The crises can be internal business conflicts or external affairs like trade policy, geopolitical issues, etc. In this regard, the organizations must develop a crisis preparedness approach that must include almost all the possible actors for a business crisis such as emergency response approach, communication crisis, IT issues and significant others. Thus, the HRD must develop a crisis management team that must create awareness among the workers concerning the adversities of an organizational crisis [9]. The recovery phase is considered the most crucial stage of crisis management. Depending on the disruptions caused by the crisis, the HRD implements a recovery approach to benefit from maintaining their wellness amidst the crisis.

The businesses of this globalized era have not encountered a severe crisis like the Covid-19 pandemic. The Covid-19 pandemic has affected both the professional and personal lives of the people [10]. The worldwide travel restrictions closed stores and restaurants, social distancing, lockdown and emerging recession have adversely affected the employers and the employees. Every organization has crisis management policies, and they train their workforce to manage the crisis period effectively and efficiently by applying specific strategic measures. However, the businesses were unprepared to deal with a global pandemic. It is expected that post-COVID-19, the policies and frameworks for crisis management in organizations are likely to change across the globe. This has created a scope to analyze the changing policies of HRD to

maintain employee welfare in a crisis period like Covid-19 without hampering organizational profitability and sustainability.

Moreover, considering the Covid-19 crisis phase, the HRD must highlight compassionate leadership skills and support the employees to retain valuable assets after the pandemic. Thus, the study highlights the correlation between HRD and crisis management in UAE, emphasising the Covid-19 crisis and the policies and framework developed by HR to manage the workers' welfare. The objectives of the article are: therefore, it has become essential to evaluate the important HRD during a crisis period and analyze the relationship between HRD and crisis management. Good crisis management can only be imposed if the policies and framework set by HRD during a crisis period like Covid-19 are highlighted in this regard, it is essential to analyze the UAE's effective measures to maintain workforce welfare during a crisis period like the Covid-19 pandemic.

Different authors have highlighted the interrelationship between HRD functions and organizational crisis management [11, 12]. However, the global pandemic is a recent event, and thus, numerous organizations have faced operational and financial backlash due to their incompetency in handling a crisis event. In this regard, particular emphasis is provided on a global crisis such as Covid-19, and how organizations can design effective policies to support their workers' welfare. HRD is not limited to a specific department of the organization. HRD responsibilities are equally distributed in every unit and department of the organization so that each employee is competent to fight the crisis scenario concerning their expertise.

This study will also define the immense HRD 's role. Organizations need to ensure that human capital is taken care of, help in the crisis management plan, and deep enrichment in crisis communication plan training in security and safety and internal talent management and related succession planning. Moreover, to the best knowledge of the researcher, this relationship between HRD and CM has never been tested in the UAE.

## Study problem

Crisis management is an essential part of every organization. This is mainly handled by the HRD of the organizations [13]. However, many organizations face failures while an organizational crisis happens, whereas some organizations emerge as successful leaders amidst the crisis period. The crisis can be internal and external; however, HRD is responsible for training and supporting the employees to face the worst situation concerning crises threats [14]. Besides training and development for organizational crisis management, there are other functions of HRD that need to be included in the crisis management plan. This consists of corporate strategy, organizational culture, organizational structure, leadership, and job value and uniqueness for corporate crisis management. On the other hand, organizations worldwide have different approaches to coping with the crisis period.

The research aims to highlight the correlation between HRD and crisis management in UAE, with particular emphasis on the Covid-19 crisis and the policies and framework developed by HR to manage the welfare of the workers.

The objectives of this research are:

- To evaluate the importance of HRD during a crisis period by examining the existing literature on it

- To examine the direct relationship between HRD dimensions and crisis management stages as described by Pearson & Mitroff, [15] model. Which will enable testing mediators and moderators of this relationship in future research.

- To analyze the effective measures taken by the UAE private and public sectors to maintain workforce welfare during a crisis period like the Covid-19 pandemic through evaluating existing journals and articles

- To highlight the policies and framework set by HRD during a crisis period like Covid-19

Contemporary businesses have faced crisis since they have started to expand beyond the domestic borders to internationalize their business operations. In this regard, the following sections review and outline the significance of HRD in organizational crises management by emphasizing various roles and responsibilities of HRD.

## Need for HRD in an organization

HRD is a model for assisting and helping employees develop their organizational and personal skills, abilities, and knowledge. HRD includes opportunities for employees such as career development, training, performance management and growth, mentoring, coaching, tuition assistance, succession planning and organizational development. The HRD concept was primarily introduced in 1969 by Leonard Nadler at a US conference [14]. He proposed HRD to be a learning experience for the organization and efficiently designed to highlight behavioural changes. Dirani et al. [16] stated that every organization needs HRD that wants to be o grow-oriented and dynamic in this fast-changing contemporary business world.

When the employees utilize their initiative, innovate, take risks and achieve targets, the organization is highlighted to possess an enabling organizational culture. Moreover, the organizations that have reached their growth limit must adapt to the changing environment. This is because no firm is immune to the requirement for processes that help obtain and maintain its capabilities for renewal and stability [17].

Therefore, HRD has different forms. It can be job shadowing or on-job training, online classes or textbook knowledge, compliance training and growth opportunities [18]. Thus, HRD is a systematic and continuous process to help the employees develop capabilities essential to perform different functions concerning their present and future organizational responsibilities. HRD is focused on enhancing supervisor and subordinate relationships so that through teamwork and collaboration, organizational efficiency is maintained. Organizational development can be improved if employees know their capabilities to bring organizational productivity.

According to Wang et al. [11], crises can lead to organizational success or failure. Considering employees' capabilities and learning difficulties when dealing with and learning from crises might lead to more success in dealing with and learning from the crisis. The authors argue that these are essential HRD concerns that can help an organization and its people survive a crisis. HRD scholars and practitioners have not paid much attention to HRD and crisis management [19]. To better grasp how learning, change, and performance-focused interventions can enhance businesses' crisis management strategies, HRD professionals may need more training in this regard.

Hutchins [12] points out that HRD activities are critical to crisis management. For HRD to be a substantial contributor to organizational crisis efforts, it must include elements other than training, organization development, and career development. In other words, HRD must have variables affecting individual learning, knowledge management, technology diffusion and use, workforce development, and organizational success. These areas align with specific emerging HRD definitions [20, 21]. Individual, group, and organizational learning and memory can assist change by aligning learning systems and processes with corporate strategic aims.

## Organizational crisis management

Crisis management is defined as the understanding and identification of significant threats to the organizations and their stakeholders, and the procedures followed by the organizations to eliminate such threats [13]. As global events are unpredictable, organizations must develop efficient strategies to cope with the drastic changes brought in the business path. Crisis management highlights the efficacy of the decision taken by an organization during a crisis period within a short time. Therefore, a crisis management plan must be developed by every organization to decrease the uncertainty in a crisis event. Bradshaw [22] stated that every business, whether small or medium, has encountered a crisis that has a severe and negative impact on normal business operations. Crises may include fire outbreaks, accidental death of an employee, CEO resigns, data breach, natural disaster or a global pandemic like Covid-19. These can lead to intangible and tangible costs to a firm concerning the loss of sales and customers, thereby decreasing the company's net income.

Responding to the aftermath of a crisis, most organizations have developed a business continuity plan in this contemporary era. The primary step of this plan is conducting a risk analysis of business operations. Risk analysis helps a business identify the potential risks existing within the current business operations or the future likelihood of its occurrence. This includes evaluating random variables and running simulations along with risk models. The risk manager can analyze the probability of risks in the coming days. Additionally, through risk analysis, the organization can handle a sudden chance that can hurt the business's functioning in the long run [23]. Once the threat is identified, the crisis management team develops a crisis plan to implement when the crisis becomes a reality.

## Effect of training on organizational crisis management

HR plays a pivotal role in crisis management by implementing effective planning and training to certify whether employees can navigate turbulent times. In the UAE, significant emergencies possess significant challenges for emergency firms and services to mitigate them once they occur. In UAE, major emergencies arise at times and locations that enhance the complexity of the events. This leads to disruption of existing resources to respond to the trouble or crisis encountered by an organization. Dubai and Abu Dhabi have faced numerous crises that have altered the organizations' crises involving natural calamities, political affairs, and others. However, both p and private organizations have managed to eradicate the crisis developed by significant emergencies.

Moreover, a lack of planning to learn from past experiences is also evident among the business of the UAE associated with significant emergency services. Thus, emergency organizations must coordinate and understand the strategic approach for actual emergencies. Hence, a comprehensive planning approach organizations to eliminate the unfolding characteristics of crises.

The contingency Planning approach is adopted by the HRD of emergency organizations so that employees are trained to work as a team, thereby, maintaining the quality of the emergency services. Thus, Alshamsi [24] highlighted the need for advanced guidelines for the UAE governing authorities to manage significant emergencies considering contingency planning. The HRD of the emergency organizations must prioritize education and training so that contingency planning is effectively implemented and monitored throughout an emergency or crisis.

## Effect of leadership on organizational crisis management

Crisis management needs the establish the ment of the team and systematic decision-making so that the units are competent and capable of planning and applying strategic decisions to

obtain successful results at the time of crisis [25]. Thus, leaders are responsible for enforcing such decisions to achieve total quality management for the sustainable improvement of the organization. The most significant role of the leader in transferring to overall quality management is to remind subordinates that quality can be achieved if efforts are imposed from the bottom to the top level of the employee base organization generates quality outcomes at the time of during at case. The leaders must monitor and support the employees at each stage of their professional and personal development. The leader must act as the role model for the employees so that the employees are encouraged and motivated to participate in quality improvement training to maintain organizational efficiency at the time of crisis.

The public sector in the UAE has decided to re-engineer its crisis management strategy, thereby introducing Business Continuity Management (BCM) as a pivotal part of the business improvement strategy. The public sectors of the UAE undertake crisis management to decrease the impact on communities from exposure to crises concerning real-world incidents [26]. The crisis leaders of such organizations have a crucial influence on BCM that generates a clear concept of this relationship. BCM practices must be pursued by the public-sector organizations in their activities and structures. This aligns with the organizational security, resilience and risk management agendas emerging from the stakeholders. The UAE public sectors believe that BCM has a significant impact on crisis performance. The organizations can be made risk-free, and the following continuity can increase their collaboration. Thus, crisis leaders must be competent to adapt, monitor and practice BCM efficiently and effectively.

The educational institutes of the UAE widely accept the idea of transformational leadership. This is because it allows immense flex and contextualized and contemporary models for Islamic culture. The introduction of the transformational leadership approach in schools of Dubai and other parts of the UAE is believed to promote and encourage change and innovation [27]. Moreover, it will help retain and respect the organization's organization nation. School initiative and culture are negatively affected if rigid rules and sudden reforms are applied. Thus, leadership must be distributed equally among all levels of education. The teachers must support this, and they need to be motivated and empowered to involve in the change agenda. The transforming leadership style is not limited to the school Principal but is also extended to the teachers to modify organizational culture gradually. It highlights that all institutional members must actively participate in the change management during or after a crisis. In this regard, the tourism industry of the UAE has developed an effective leadership strategy that identifies the significance of sustainable development amidst the crisis. Thus, an efficient association is evident between strategic leadership, strategic planning efficacy and tourism organization competitiveness in the UAE tourism industry.

## Effect of organizational strategy on organizational crisis management

Strategic planning organizations handle their threats and weaknesses during a crisis [28]. Thus, the organizational strategy is vital to managing the corporate crisis. Vardarlıer [29] stated four strategies to deal with a crisis event. The prime one is the escape strategy, which is considered harmful during a crisis where the leaders blame staff performance as the core factor of the crisis. This is followed by a containment strategy where the leaders are focused on reducing the effect of the crisis, adopting the limiting and freezing concept. The most efficient method is the teamwork or cooperation strategy where the employees work to cope with the crisis event.

Alzaabi et al. [30] stated that the UAE civil defence performance is a crucial issue during a crisis in Dubai and other regions of the UAE. In this regard, strategy management and a

balanced scorecard can impact the organizational performance of civil defence organizations. Through this approach, the civil defence departments will help achieve the country's objectives and mission during a crisis event. Although the developing countries of the West develop this approach, it can be helpful for the public organizations of Dubai and the rest of the UAE involved in general and civic defence.

## Effect of organizational structure on organizational crisis management

The organizational structure is efficient in tackling the threats during a crisis period. However, adaptation to a new organizational structure and design can help it effectively understand the latest crisis [31]. Under crisis operations, strategic plans and corporate performance become significant for an organization. Organizations can change their structure through decision-making and training procedures when faced with crises. This restructuring is usually towards a complicated organizational form with enhanced flexibility in accessing the resources. The training provided by the organization to its employees determines whether the organization will benefit from the restructuring during a crisis period. The phenomena of organizational restructuring are interesting as it highlights that during a crisis, companies tend to expand their collaborations to have enhanced control of their resources rather than becoming leaner. During a crisis event, the organizations must improve their resource access to reduce the crises even if the organizational restructuring enhanced their structural redundancy.

In response se to the Covid-19 pandemic, the UAE government authorities have designed advanced guidance and legislation to assist private sector employers in handling it. The flexible working arrangement is a necessary organizational restructuring implemented by the private sector employers of Dubai. Through this restructuring, the businesses can support the employees and maintain organizational efficiency simultaneously. The MoHRE (Ministry of Human Resources and Emiratisation) has introduced new resolutions for companies maintain employees' health and well-being [32].

## Effect of organizational culture on organizational crisis management

A reciprocal causal relationship is evident between the organizational crisis and culture [33]. If the corporate culture is ineffective and the animation of an organization in crisis is cunningly interrupted. If the organizational structure is unhealthy, a crisis can highlight the hidden core values behind the demonstrated corporate culture. Moreover, crises can initiate from an organizational culture and may expand to the administrative subsystem without outlining its path unless severe. Thus, organizational culture must be competent in managing a crisis faced by an organization.

Organizational culture is influential in identifying and segregating between crisis-prone and crisis-prepared. On the contrary, administrative on corporate crisis management is lower, and the existing literature on crisis studies is fragmented. Abo-Murad et al. [34] highlighted this fragmentation and segregated crisis to a linear paradigm that evaluates the crises and denies their dynamic nature.

Crisis communication is considered an essential factor for UAE organizations to develop a crisis communication plan and strategies by involving a network of stakeholders in response to a crisis. During a crisis, the UAE organizational culture supports a three-stage model that includes a detailed approach to handle the risks associated with the crisis event [35]. This consists of a pre-crisis, response and post-crisis plan. The UAE organizational culture is different from the West, and this culture influences the management system. The difference lies in the

use of religion, language, the limited number of female employees, and the involvement of masculinity in all organizational decisions. Organizations maintain effective crisis communication to reduce risks during a crisis event. Both public and private organizations in the Dubai nations strive to maintain effective teamwork and collaboration so that employee performance and organizational efficiency are affected.

## Effect of job values and uniqueness on organizational crisis management

Employee satisfaction is the main reason for organizational efficiency. If employees are valued and treated with respect, the organization can manage the turnover rate. Corporatization leaders must encourage diverse contributions and help the subordinates contribute to the decision-making process fully. Besides focusing on collective goals, specific behaviors of the leaders towards the employees may enhance belongingness and highlight value for uniqueness [36]. A climate of inclusion during the crisis is valuable to maintaining organizational reputation. The leaders must include employees' needs and preferences while designing a crisis management plan. This will ensure effective management of the corporate crisis.

Employee happiness is essential, as evident from the private healthcare organizations of Dubai [37]. The secret of happiness at work may improve crisis management issues. The organization's capability to maintain employee well-being during a crisis in acquiring long-term employee loyalty. Employee satisfaction in the Dubai private healthcare sector includes a significant relationship with colleagues and supervisors, appraisal, work-life balance, job security, and career development opportunities. Critical events can be mitigated efficiently if healthcare workers are happy to work for the organization.

## Literature gap

This literature widely explores the significance of HRD practices in the UAE. This includes the importance of training, organizational culture, structure, strategy, and job values for corporate crises management in Dubai, UAE. However, the literature also suggests the influence of language, culture, religion, and ethnic values in the UAE associated with HRD functions. This highlights that HRD functions and effective HR planning in organizational crisis management can be hampered due to Arab culture and traditional beliefs. This creates a literature gap and instigates further investigation in comparing Western and Arab cultures to highlight the efficacy of HRD in organizational crisis management. In short, the literature gap states that although crisis management is efficiently tackled in the UAE private and public sectors, the Arab culture and the organizational diversity due to globalization may create barriers in such implementation. It is evident in the paper that Arab culture does not permit certain practices, such as the inclusion of women in organizational responsibilities and others. Therefore, a lack of adequate knowledge to tackle a crisis by a female worker may have a disastrous effect on the organization.

Previous studies have recognized the importance of crisis management activities in organization management functions as the impact of the crisis on individual and organizational performance has been increasingly recognized [11]. However, the existing theories and literature have been found to focus on management itself but ignore the main component of the crisis, human resources. Even in studies that discuss a human resource crisis, human resources' role in dealing with such problems is neglected. Therefore, this study explores how human resource development can achieve organizational capacity and learning, which will enhance the organization's resilience, confidence, and ability to deal with unexpected events as soon as they arise.

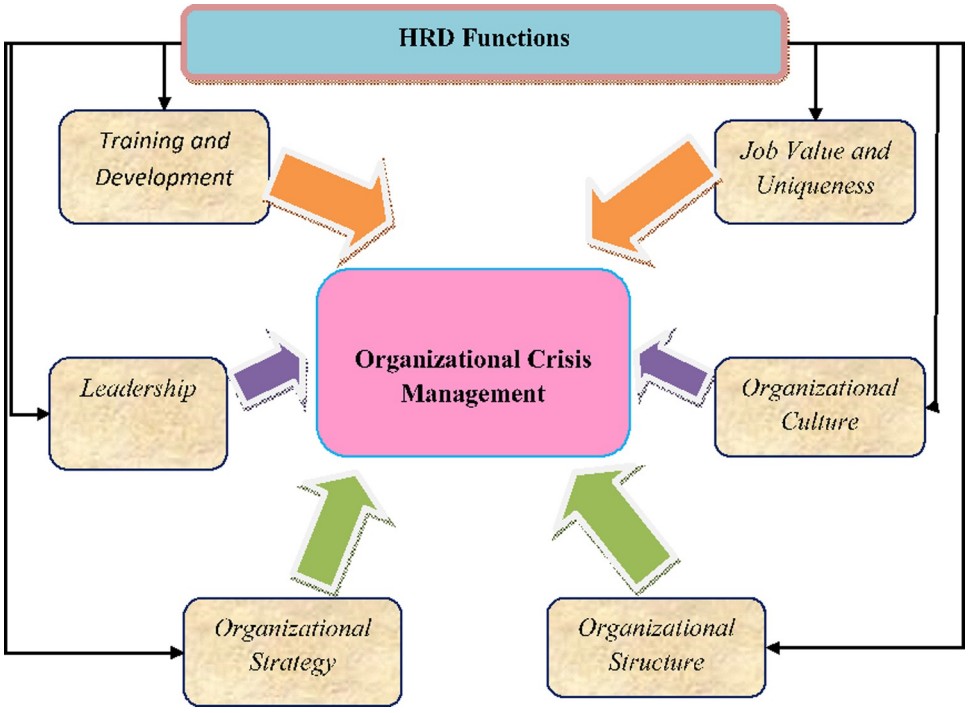

**Fig 1. Conceptual framework.**

This paper aims to explore the role of human resource development in managing crises in organizations.

## The methodology

The proposed research methodology is quantitative. The model proposed in Fig 1 will be tested using the path modeling techniques. According to Calantone et al. [38] "The Partial Least Squares (PLS) Path modeling parameter estimates better reveal the strength and direction (i.e., positive vs. negative) of the relationships among variables compared to correlation coefficients"(p.28), "PLS avoids parameters estimation biases common in regression analysis". The core technique utilized to generate data to answer the above research aims will be a survey. A questionnaire was prepared to measure all the study variables, reliability and validity was tested on a pilot study consisting of 20 respondents. The pilot study also was used to gauge the consistency of the questionnaire. All suggestions and comments expressed by the experts and the respondents were included in the final assessment. Before the survey, five HRD and Crisis management experts were engaged in reviewing the content readability, clarity, and comprehension of the questionnaire for the study.

All, the measurement items in the questionnaires were measured using a five-point Likert scale ranging from 1 representing "Strongly Disagree" to 5 representing "Strongly Agree". In developing the instrument, measurement items were selected and adapted from validated questionnaires used in previous researches.

To check the hypotheses, we targeted employees working with government institutions in Dubai to assess how HRD influenced their crisis management during COVID 19 Crisis. All employees were selected as it is assumed that usually all employees will be involved somehow with the crises or with one of its dimensions.

**Table 1. Distribution of sample and population.**

| Name of Institution | Total number of employees working in Dubai | Sample |
|---|---|---|
| Ministry of Interior | 5000 Employee | 75 |
| Ministry of Education | 7000 Employee | 105 |
| National Emergency Crisis and Disasters Management Authority | 3000 Employee | 45 |
| General Civil Aviation Authority | 850 employees | 15 |
| General Authority of Islamic Affairs & Endowments | 1000 Employee | 15 |
| Emirates Telecommunication Establishment (Etisalat) | 3000 Employee | 45 |
| Total | 19850 | 300 |

Table 1 below shows the ministries and the approximated total number of employees working in those entities, and the sample size for each Institution.

Table 2 below shows each variable's statements and the sources used for each account.

**Table 2. Sources of statements used in the survey.**

| Dependent Variable | Source |
|---|---|
| **Early warning signs** | |
| 1. The organization's management monitors any signs of weaknesses or Problems that indicate a crisis. | [39] |
| 2. The organization's management monitors malfunctions or disturbances that indicate a crisis. | [40] |
| 3. The staff is qualified to perform the skills of collecting and analyzing indicators of the crisis. | [40] |
| **Preparation and Prevention Stage** | |
| 1. The organization incorporates crisis readiness into the strategic plan | [41] |
| 2. The organization evaluates the progress of the strategic plan | [42] |
| **Damage Containment Stage** | |
| 1. The organization prevents the spreading of the damage to the uncontaminated parts of the organization | [43] |
| 2. The organization ensures accurate crisis management by testing crisis management capabilities before the crisis occurs | [44] |
| **Activity Recovery Phase** | |
| 1. The organization implements long-term and short-term business recovery programs to facilitate the continuation of the business operations | [45] |
| 2. The organization designs such programs based on the essential services to resume business operations. | [46] |
| **Learning Stage** | |
| 1. The organization learns the new ways of preventing the crisis event | [47] |
| 2. The organization analyses and compares the factors responsible for effective crisis management | [46] |
| **HRD independent variables** | |
| *training* | |
| 1. The workers are trained to evaluate the early signs of crisis | [48] |
| 2. The employees are provided with coaching on how to tackle the risks during a crisis period | [48] |
| 3. The employees are trained for effective communication within the team | [48] |
| 4. The employees develop a plan and demonstrate it if a crisis occurs | [48] |
| *Leadership* | |
| 1. The leaders design the training process for crisis management | [25] |
| 2. The leaders implement an adequate strategy to cope with the organizational crisis | [25] |
| 3. The leadership strategy is motivational. | [25] |
| 4. The leaders are empathetic towards the employees | [25] |
| 5. The leaders evaluate the progress of the crisis management plan | [25] |

(*Continued*)

**Table 2.** (Continued)

| *organizational strategy* | |
|---|---|
| 1. The organization understands the intensity of the crisis | [29] |
| 2. The organization assigns definite roles to the employees | [29] |
| 3. The organization clarifies the purpose of different strategies implemented during the crisis to its teams | [29] |
| 4. The organizational strategy is beneficial for the employees and the organization too | [29] |
| *organizational structure* | |
| 1. The organization implements line organizational structure at the time of crisis management | [31] |
| 2. The line organizational structure is effective as the authority will flow from top to bottom | [31] |
| 3. The organization experiences effective communication among the team members through this organizational structure | [31] |
| 4. The organization has enhanced control on the resources | [31] |
| *Job Values and uniqueness* | |
| 1. The organization values its employees during the crisis period | [36] |
| 2. The organizational leaders are respectful towards the workforce | [36] |
| 3. Employee satisfaction enhances organizational productivity during the crisis period | [36] |
| 4. Job value and uniqueness are major contributing factors for organizational crisis management | [36] |
| *Organizational Culture* | |
| 1. The organization promotes a culture of continuous improvement in crisis management. | [41] |
| 2. The organization believes that the increased acceptance of crisis management among employees is essential. | [49] |
| 3. The organization employs a system of norms and beliefs that support the receiving, interpreting, and translating signals of Crisis. | [50] |
| 4. The organization's members widely understand the values and symbols that form the organization's beliefs. | [50] |

After data collection and to test the direct causality between independent variables and the dependent variable, a Structural Equation Model (SEM) using Smart PLS software will be used. SEM, sometimes called the Path analysis, is used for complicated Models and where latent variables are used.

## Hypothesis development

*Hypothesis 1 There is a direct effect of training on Organizational Crisis Management*

Training is necessary to empower the employees to tackle a crisis event in an organization as it will prepare, educate and train the workforce on how to manage a crisis. A crisis management program starts with a professional team, which acts as the organizational backbone for crisis preparation and response. In practice, system failures and poor coordination can cause serious errors. Crisis management training teaches leadership, communication skills and increases team performance [51].

*Hypothesis 2 There is a direct effect of leadership on Organizational Crisis Management*

A practical leadership approach is necessary for a systemic decision-making process during a crisis period. Leaders must motivate the employees to acquire total quality management during an organizational crisis. Therefore, employees must be included in the decision-making process of the organization. The leaders must act as role models for the employees.

*Hypothesis 3 There is a direct effect of organizational strategy on Organizational Crisis Management*

Strategic planning helps an organization to understand the strengths and loopholes it possesses to tackle the crisis. Teamwork or cooperation is the most efficient strategy an

organization adopts during a crisis period. Leadership is crucial in this case, and the leaders, along with the subordinates, confront the crisis event in a scientific approach.

*Hypothesis 4 There is a direct effect of organizational structure on Organizational Crisis Management*

The organizational structure affects organizational productivity during the time an organization is likely to change organizational structure during a short time to quickly recover from the crisis. A strategic design, such as crisis training and empowerment, will help effective organizational performance.

*Hypothesis 5 There is a direct effect of organizational culture on Organizational Crisis Management*

Organizational culture has a direct influence during a corporate crisis. If the corporate culture is ineffective, teamwork is less evident, and administrative efficiency is not obtained. Therefore, corporate culture and crisis management must be correlated. This will help the organization to manage the corporate crisis accurately.

*Hypothesis 6 There is a direct effect of Job Values and uniqueness on Organizational Crisis Management*

Employee satisfaction can be achieved if employees are valued and respected. Organizational crisis primarily affects the employees, which are followed by corporate profits. In this case, if the employees are not valued, and job uniqueness is not created, the corporate crisis will take a long time to be mitigated.

## Assessment of measurement model

To test the convergent validity, three factors extracted are Factor loading, Composite reliability, and average variance, as shown in Table 3. All the loading values within the table are above the cut-off point (0.7), which means that there is no issue with these data according to the commonality test. Composite reliability indicators were higher than 0.7 for all the constructs. The average of variance extracted (AVE) was also examined for each construct, and values were substantially higher than Chin's [52] suggested 0.5 thresholds.

Regarding the discriminant validity cross-loading Variable Correlation- Root square comparison was conducted in Tables 4 and 5. Table 4 shows that the square root of AVE for each construct was higher than the inter-scale correlation. Table 5 shows that all items loaded with a higher value with the same construct than other variables. This comparison satisfies the discriminant validity suggested by Chin's [53] criteria. In summary, these results indicate satisfactory reliability and convergent validity.

## Structural model and hypothesis testing

Using SEM-PLS, we used the following criteria to assess the hypothesis model: $R^2$ adjusted value, Beta Coefficient, and $f^2$ effect size. Before testing the structural model, fit adjustment with Standardized Root Mean Square Residual value was evaluated. The result was 0.06, which indicated a good fit adjustment.

In respect to the predictive power of the model provided for Crisis Management, $R^2$ adjusted value indicates that the model explains 45% of the variance in CM. Bootstrapping was performed to provide a significance level for each hypothesized relationship, parameter settings for bootstrapping included no sign changes, and the 500 samples. According to the results, Training, Leadership, Organisation strategy, and Organisation culture predict significantly CM to use online methods to shop (see Table 5), in particular, Training showed to be the best predictor, followed by the Organisation culture. According to $f^2$, values of 0.02, 0.15, and 0.35 represent small, medium, and large effect sizes, respectively. The $f^2$ effect size on the

**Table 3. Results of measurements model-convergent validity.**

| Construct | Items | Loading | CR | AVE |
|---|---|---|---|---|
| Training | T1 | 0.854 | 0.916 | 0.731 |
| | T2 | 0.876 | | |
| | T3 | 0.854 | | |
| | T4 | 0.836 | | |
| Leadership | L1 | 0.854 | 0.903 | 0.655 |
| | L2 | 0.712 | | |
| | L3 | 0.903 | | |
| | L4 | 0.766 | | |
| | L5 | 0.882 | | |
| organizational strategy | OS1 | 0.895 | 0.882 | 0.662 |
| | OS2 | 0.792 | | |
| | OS3 | 0.909 | | |
| | OS4 | 0.882 | | |
| Organizational structure | OST1 | 0.725 | 0.867 | 0.629 |
| | OST2 | 0.899 | | |
| | OST3 | 0.902 | | |
| | OST4 | 0.785 | | |
| Job Values and uniqueness | JVU1 | 0.753 | 0.905 | 0.704 |
| | JVU2 | 0.84 | | |
| | JVU3 | 0.884 | | |
| | JVU4 | 0.873 | | |
| Organizational Culture | OC1 | 0.792 | 0.877 | 0.649 |
| | OC2 | 0.918 | | |
| | OC3 | 0.91 | | |
| | OC4 | 0.745 | | |
| Crisis Management | ARP1 | 0.819 | | |
| | ARP2 | 0.823 | 0.933 | 0.566 |
| | DCS1 | 0.806 | | |
| | DCS2 | 0.798 | | |
| | EWS1 | 0.836 | | |
| | EWS2 | 0.821 | | |
| | EWS3 | 0.74 | | |
| | LS1 | 0.822 | | |
| | LS2 | 0.818 | | |
| | PPS1 | 0.773 | | |
| | PPS2 | 0.815 | | |

CM was small for all the variables, which implies a small but significant contribution of the variables whose hypotheses were confirmed. All these results are summarized in Table 6. Path analysis results are addressed in Fig 2.

## Discussion of results

In this study, Human resource development for crisis management activities in the public sector in Dubai was discussed. According to Alshamsi [24], the UAE lacks effective coordination within its crisis management teams. In this regard some country institutions such as the UAE and Dubai tourism industry have developed a conscious and successful management strategy so that the sector can handle any crisis event [54].

**Table 4. Square root of AVE.**

| | Crisis Management | Job values and uniqueness | Leadership | Organization Culture | Organization strategy | Organization structure | Training |
|---|---|---|---|---|---|---|---|
| Crisis Management | **0.752** | | | | | | |
| Job values and uniqueness | 0.537 | **0.839** | | | | | |
| Leadership | 0.700 | 0.646 | **0.809** | | | | |
| Organization Culture | 0.501 | 0.826 | 0.621 | **0.806** | | | |
| Organization strategy | 0.582 | 0.788 | 0.741 | 0.755 | **0.814** | | |
| Organization structure | 0.513 | 0.722 | 0.645 | 0.791 | 0.715 | **0.793** | |
| Training | 0.744 | 0.584 | 0.752 | 0.563 | 0.602 | 0.562 | **0.855** |

Note: Bold values indicate higher factorial loads.

Hawarna et al. [55] highlighted that DQA (Dubai Quality Award) is a significant organizational strategy to enhance organizational performance among the private organisations of Dubai. Additionally, it is an important quality award that has been applied across the private sector of the UAE to improve organizational performance. To identify organizational efficiency during a crisis period in Dubai, it is essential to understand the importance of DQA and its positive influence on HRD practices. A positive impact of DQA suggests that companies across Dubai have efficient HRD practices. Thus, organizational efficiency and performance are achieved by the organisations during a crisis period. HRD acts as a strategically important mediator between DQA and organizational performance in the private sectors of Dubai.

A crisis model for a group of government entities that are involved with the crisis was suggested. Concerning the fact that test results of hypothesis four out of six HRD functions were found to have a direct effect on CM, it can be said that training, leadership, organizational strategy and organizational culture have a positive and significant impact on crisis management effectiveness. It means that upon an increase of those four factors, crisis management effectiveness can be increased.

The organizational structure has no direct impact on enhancing CM within the content of the Dubai public sector. Collaboration relationships are complex, particularly during times of organizational crisis; their structuring principles vary according to the organizational structure and other factors such as the organization's mission, the stage of the collaboration process, and the member's role within the organization. The organizational structure facilitates the flow of information between all its levels. The flow of information easily is necessary for success in crisis management. The organizational structures of government entities in Dubai are rigid which doesn't allow collaboration and smooth flow of information during crises. On the other hand, as Hawara et al. [55] report, government parallel structures, such as the DQA (Dubai Quality Award) represent a significant organizational strategy to enhance organizational performance among the private organizations of Dubai. Additionally, it is an important quality award that has been applied across the private sector of the UAE to improve organizational performance. To identify organizational efficiency during a crisis period in Dubai, it is essential to understand the importance of DQA and its positive influence on HRD practices. A positive impact of DQA suggests that companies across Dubai have efficient HRD practices. Thus, organizational efficiency and performance are achieved by the organizations during a crisis period. HRD acts as a strategically important mediator between DQA and organizational performance in the private sectors of Dubai. This is aligned with the results of the study on the role of leadership on crisis management in the UAE.

**Table 5. Discriminant validity- cross loading.**

|  | Crisis Management | Job values and uniqueness | Leadership | Organization Culture | Organization strategy | Organization structure | Training |
|---|---|---|---|---|---|---|---|
| ARP1 | **0.819** | 0.447 | 0.601 | 0.431 | 0.488 | 0.408 | 0.64 |
| ARP2 | **0.823** | 0.442 | 0.589 | 0.41 | 0.491 | 0.391 | 0.644 |
| DCS1 | **0.806** | 0.457 | 0.566 | 0.441 | 0.489 | 0.441 | 0.583 |
| DCS2 | **0.798** | 0.449 | 0.552 | 0.402 | 0.459 | 0.409 | 0.629 |
| EWS1 | **0.436** | 0.237 | 0.241 | 0.187 | 0.281 | 0.244 | 0.291 |
| EWS2 | **0.621** | 0.303 | 0.366 | 0.249 | 0.313 | 0.287 | 0.418 |
| EWS3 | **0.64** | 0.291 | 0.369 | 0.24 | 0.319 | 0.264 | 0.449 |
| PPS1 | **0.773** | 0.363 | 0.497 | 0.346 | 0.391 | 0.379 | 0.527 |
| PPS2 | **0.815** | 0.375 | 0.543 | 0.359 | 0.465 | 0.403 | 0.543 |
| LS1 | **0.822** | 0.49 | 0.634 | 0.477 | 0.507 | 0.448 | 0.695 |
| LS2 | **0.818** | 0.48 | 0.652 | 0.467 | 0.516 | 0.488 | 0.729 |
| JVU1 | 0.392 | **0.753** | 0.452 | 0.58 | 0.586 | 0.629 | 0.414 |
| JVU2 | 0.483 | **0.84** | 0.574 | 0.654 | 0.657 | 0.658 | 0.495 |
| JVU3 | 0.463 | **0.884** | 0.539 | 0.754 | 0.675 | 0.729 | 0.522 |
| JVU4 | 0.456 | **0.873** | 0.592 | 0.773 | 0.72 | 0.742 | 0.522 |
| L1 | 0.631 | 0.544 | **0.854** | 0.513 | 0.603 | 0.514 | 0.718 |
| L2 | 0.475 | 0.466 | **0.712** | 0.478 | 0.529 | 0.501 | 0.508 |
| L3 | 0.63 | 0.601 | **0.903** | 0.583 | 0.66 | 0.608 | 0.674 |
| L4 | 0.451 | 0.344 | **0.666** | 0.307 | 0.439 | 0.333 | 0.478 |
| L5 | 0.617 | 0.617 | **0.882** | 0.593 | 0.732 | 0.619 | 0.63 |
| OC1 | 0.36 | 0.583 | 0.446 | **0.792** | 0.559 | 0.552 | 0.397 |
| OC2 | 0.513 | 0.781 | 0.594 | **0.918** | 0.705 | 0.743 | 0.562 |
| OC3 | 0.465 | 0.782 | 0.594 | **0.91** | 0.706 | 0.747 | 0.528 |
| OC4 | 0.109 | 0.445 | 0.256 | **0.745** | 0.408 | 0.48 | 0.188 |
| OS1 | 0.546 | 0.691 | 0.685 | 0.684 | **0.895** | 0.719 | 0.564 |
| OS2 | 0.172 | 0.359 | 0.322 | 0.338 | **0.792** | 0.427 | 0.181 |
| OS3 | 0.517 | 0.717 | 0.667 | 0.671 | **0.909** | 0.724 | 0.548 |
| OS4 | 0.538 | 0.717 | 0.651 | 0.684 | **0.882** | 0.737 | 0.541 |
| OST1 | 0.188 | 0.367 | 0.278 | 0.383 | 0.431 | **0.725** | 0.252 |
| OST2 | 0.495 | 0.748 | 0.595 | 0.721 | 0.775 | **0.899** | 0.527 |
| OST3 | 0.483 | 0.801 | 0.606 | 0.739 | 0.75 | **0.902** | 0.508 |
| OST4 | 0.374 | 0.589 | 0.488 | 0.59 | 0.562 | **0.785** | 0.435 |
| T1 | 0.643 | 0.474 | 0.624 | 0.464 | 0.492 | 0.483 | **0.854** |
| T2 | 0.648 | 0.49 | 0.638 | 0.474 | 0.53 | 0.475 | **0.876** |
| T3 | 0.678 | 0.536 | 0.639 | 0.475 | 0.522 | 0.479 | **0.854** |
| T4 | 0.644 | 0.497 | 0.673 | 0.513 | 0.513 | 0.488 | **0.836** |

Note: Bold values indicate higher factorial loads.

Abo-Murad et al. [34] stated that organizational culture helps in strengthening the crises preparedness of a company. The results of the present study corroborate these findings for the UAE. Middle east countries rely heavily on tradition, believes and customs. They provide a sense of security, preparedness and resilience that nurtures from experience. In that sense, culture might have a stronger role in certain cultures that share similar traits. Crisis management can be correlated with resilience. Williams et al. [56] revealed that resilience highlights the means of strategic misalignments, as well as adjusting and responding to triggered events. By combining resilience and crisis, it is possible to understand the success or failure level of an

**Table 6. Path coefficient of the research hypotheses.**

|  | Std. Beta | Std. Error | T-value | P- Value | Decision |
|---|---|---|---|---|---|
| Training -> Crisis Management | 0.540 | 0.052 | 10.319 | 0.000 | Supported** |
| Leadership -> Crisis Management | 0.231 | 0.065 | 3.543 | 0.000 | Supported** |
| Organisation strategy -> Crisis Management | 0.205 | 0.068 | 3.748 | 0.031 | Supported* |
| Organisation structure -> Crisis Management | 0.036 | 0.069 | 0.518 | 0.605 | Not Supported |
| Organisation Culture -> Crisis Management | 0.239 | 0.116 | 2.625 | 0.026 | Supported* |
| Job values and uniqueness -> Crisis Management | 0.051 | 0.062 | 0.815 | 0.416 | Not Supported |

Significant at P** = < 0.01, p* = <0.05.

organization. In this case, resilience arising from experience might be rooted on culture. A future line of research could provide further insight on the mediation effect of resilience on crisis management.

Different levels of values and the uniqueness of jobs may theoretically influence how HRD responds to crisis management [57]. Numerous job roles are critical in crisis management. Bulmahn and Krakel [58] emphasized that key organizational leaders' primary responsibilities include focusing on information exchange to ensure coherence before the onset of a crisis and preparing for unknown crises. In the context of Dubai public entities, it looks that Values and Uniqueness do not affect CM. Further research is needed to explain why.

As the direct effect of HRD has been proved, further research is needed to explain this direct impact. This can be done by expanding the Model by introducing mediators and Moderators to explain this direct impact.

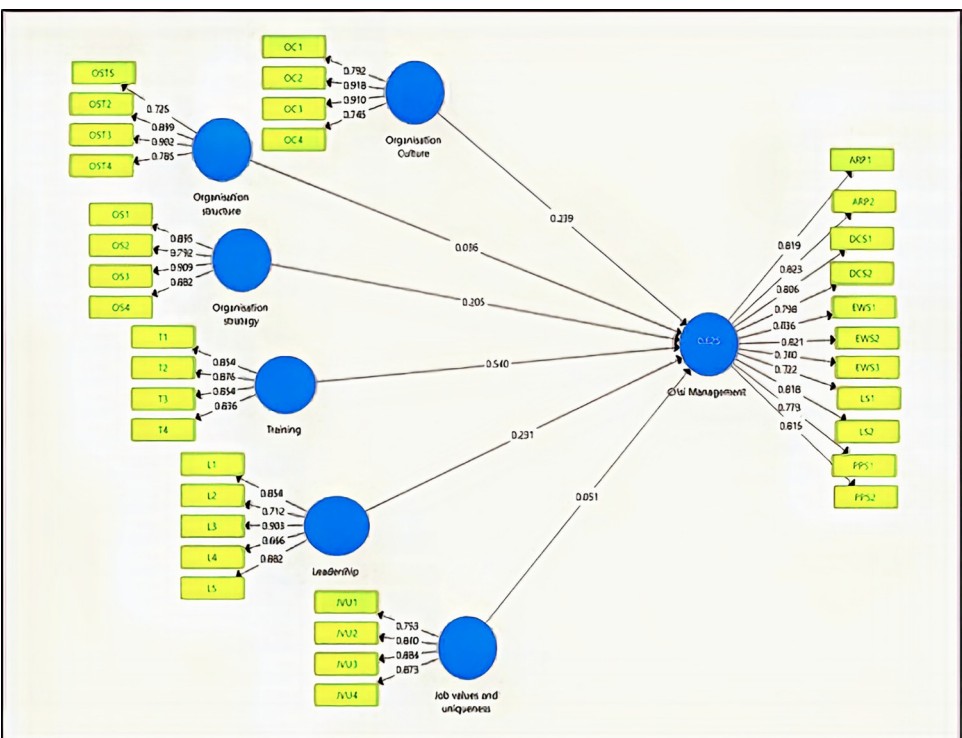

**Fig 2. Path analysis results.**

Finally, Crisis management theories should broaden their current boundaries to include the HRD contribution. The study model suggests a unique perspective on the capability and learning issues pertinent to crisis management. It does not, however, address how these activities are carried out. As a result, the next step in developing HRD's contribution is to conduct a more in-depth examination of the resulting political and change implementation issues.

## Conclusion

In this study, particular emphasis is made in Dubai, UAE to explore the HRD functions during crisis management and to discuss the HRD functions in the private and public sectors of the UAE. The data findings suggest that although HRD is well-developed in the UAE organizations, certain loopholes such as lack of effective teamwork and accurate crises management plan create barriers in its successful implementation.

Training has a positive and significant influence on crisis management. In other words, with an increase in employee training, there is better crisis management.

Leadership positively and significantly influences crisis management. That is, if the leaders of the organization carry out their task in a way that is respected and the employees follow their guidelines willingly, it leads to better crisis management.

Organizational strategy positively and significantly influences better crisis management. That is, if the organizational management is well defined, it is clearly transmitted to the employees, it generates good crisis management.

The organizational structure does not significantly influence crisis management.

Organizational culture positively and significantly influences crisis management. In other words, a well-defined culture, consolidated among employees and well transmitted, produces better crisis management.

Values and uniqueness of work do not significantly influence crisis management

The main findings of the study reflect that training, leadership, organizational strategy, and organizational culture directly positively impact the efficiency of Crisis management (CM) during the Covide-19 crisis in the public entities of Dubai-UAE. In particular, training showed to be the best predictor, followed by Organizational culture.

## Supporting information

**S1 Data.**
(XLSX)

## Author Contributions

**Conceptualization:** Amir Hamad Salim Binnashira Alketbi.

**Data curation:** Amir Hamad Salim Binnashira Alketbi.

**Formal analysis:** Amir Hamad Salim Binnashira Alketbi.

**Project administration:** Juan Antonio Jimber del Rio.

**Supervision:** Juan Antonio Jimber del Rio.

**Validation:** Alberto Ibáñez Fernández.

**Visualization:** Alberto Ibáñez Fernández.

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
