## [Decision Letter · Decision Letter 0]

7 Dec 2021

PONE-D-21-34065Exploring the role of Human Resource Development functions on Crisis management: the case of Dubai-UAE during Covid-19 crisisPLOS ONE

Dear Dr. JUAN ANTONIO Jimber del Río,

Thank you for submitting your manuscript to PLOS ONE. After careful consideration, we feel that it has merit but does not fully meet PLOS ONE’s publication criteria as it currently stands. Therefore, we invite you to submit a revised version of the manuscript that addresses the points raised during the review process.

We look forward to receiving your revised manuscript.

Kind regards,

Rogis Baker, Ph.D

Academic Editor

PLOS ONE

Journal Requirements:

2. Please ensure that you refer to Figure 2 in your text as, if accepted, production will need this reference to link the reader to the figure.

3. We note you have included a table to which you do not refer in the text of your manuscript. Please ensure that you refer to Tables 1 and 2 in your text; if accepted, production will need this reference to link the reader to the Table.

4. Please include a copy of Table 4 which you refer to in your text.

Reviewers' comments:

Reviewer's Responses to Questions

**Comments to the Author**

1. Is the manuscript technically sound, and do the data support the conclusions?

Reviewer #1: Yes

Reviewer #2: Partly

2. Has the statistical analysis been performed appropriately and rigorously? 

Reviewer #1: Yes

Reviewer #2: I Don't Know

3. Have the authors made all data underlying the findings in their manuscript fully available?

Reviewer #1: Yes

Reviewer #2: Yes

4. Is the manuscript presented in an intelligible fashion and written in standard English?

Reviewer #1: Yes

Reviewer #2: No

5. Review Comments to the Author

Reviewer #1: This paper examines the role of HRD functions on the effectiveness of crisis management. For the most part, the paper is clearly written, the research idea is interesting, and the research design is impressive in many respects. Although acknowledging these many positives, I do have some major concerns with your theoretical framing and methodology and some minor concerns on trivial issues. I wish you the best with your work moving forward.

Major concerns:

1.Insufficient problematization of the research area. As the paper has put in the introduction, since “involve employee welfare as a crucial aspect of crisis management” is critical, HRD could play a significant role in the current crisis for organizations due to the Covid-19 pandemic. The authors put a lot of effort into illustrating the importance of studying the relationship between HRD functions and crisis management. Despite that there are “study problem” and “literature gap” sections in the paper, however, the elaboration on what’s the theoretical problems in this area and why your research is urgent is not enough. What have other scholars done on figuring out the role of HRD functions in crisis management? What are the flaws in these studies? What incremental or fundamental contributions the authors could make above and beyond previous wisdom?

2.Underdeveloped hypotheses development. Arguments in the hypotheses development section are generally thin and weak. For instance, in the reasoning for Hypothesis 1, the authors mainly stated the logic in the first sentence (without any citation). Specifically, the authors were literally saying that training is useful in crisis management because training would “train the workforce on how to manage a crisis.” How exactly training works in crisis management due to Covid-19? For the five stages of your dependent variable, which would be affected most by training practice? And the same with every hypothesis else. The authors need a more in-depth reasoning process to develop the hypotheses.

3.Model configuration. As the authors distinguished between different stages of crisis management, why not test them separately in the analysis and get a derived finding from the basic model? I know that may make the model complex, but the results would be interesting if you can find that different HRD functions work at different stages of crisis management.

Minor issues:

1.Missing page number in REFERENCES:

Abo-Murad, M., Abdullah, A. K., & Jamil, R. (2019). Effect of the Organisational Culture on

Crisis Management in Hotel Industry: A Qualitative Exploration. International Journal

of Entrepreneurship, 23(2).

Otoo, F. N. K., & Mishra, M. (2018). Measuring the impact of human resource development

(HRD) practices on employee performance in small and medium scale

enterprises. European Journal of Training and Development.

https://doi.org/10.1108/EJTD-07-2017-0061

Vardarlıer, P. (2016). Strategic approach to human resources management during

crisis. Procedia-Social and Behavioral Sciences, 235(2).

https://doi.org/10.1016/j.sbspro.2016.11.057

2.Fonts are not unified.

3.Some period (s) were missing. Such as “This is because no firm is immune to the requirement for processes that help to obtain and maintain its capabilities for renewal and stability (Armitage, 2018)”

I hope you find my comments useful. Good luck with your research!

Reviewer #2: Appreciating your work, please find hereunder some comments:

1. The overall format of the article does not follow PLOS guidelines, for instance line numbers and headings that make it difficult to give feedback. Additionally, it appears that it follows a format for a research proposal than an a research communication manuscript.

N.B. While the English is readable, I would also suggest to have the overall grammatical flow reviewed by a professional in that regard.

2. Abstract lacks a background segment; as it stands, it is a summary of the methods and results. Also, HRD should only be abbreviated after first being mentioned in full capacity.

3. Consider moving third paragraph of the introduction to second to last as the flow correlates more there.

4. Consider moving last paragraph of the study problem to a separate conclusion segment.

5. Suggest to summarize aim and objectives, along with those mentioned in the introduction, into one paragraph that ties up the overall flow of the introduction segment.

6. Suggest to remove literature review as a separate segment and summarize the contents to fit into the introduction segment.

7. Suggest to broaden the discussion segment to include comparative analysis of other studies, such as those mentioned in the literature review. Matter of fact, it would make the most sense to condense the literature review to narrow down its contents to those that further strengthen the introduction and move a bulk of the literature review contents to the discussion segment for a comparative narration and analysis.

8. Suggest to have a separate conclusion and discussion segment.

9. Can the authors provide their questionnaire in a separate file?

10. Can the authors cite the legal framework/IRB protocol that waived consent in this case? I am assuming consent was waived as there is no mention of consent in the manuscript.

6. PLOS authors have the option to publish the peer review history of their article (what does this mean?). If published, this will include your full peer review and any attached files.

Reviewer #1: No

Reviewer #2: No

---

## [Author Response · Author response to Decision Letter 0]

4 Jan 2022

Reviewer 1:

1. Please ensure that you refer to Figure 2 in your text as, if accepted, production will need this reference to link the reader to the figure.

Figure 2 is referred in the text. 

2. We note you have included a table to which you do not refer in the text of your manuscript. Please ensure that you refer to Tables 1 and 2 in your text; if accepted, production will need this reference to link the reader to the Table.

Tables 1 and 2 are referred in the text.

3. Please include a copy of Table 4 which you refer to in your text.

Table 4 is included in the text.

Major concerns:

1.Insufficient problematization of the research area. As the paper has put in the introduction, since “involve employee welfare as a crucial aspect of crisis management” is critical, HRD could play a significant role in the current crisis for organizations due to the Covid-19 pandemic. The authors put a lot of effort into illustrating the importance of studying the relationship between HRD functions and crisis management. Despite that there are “study problem” and “literature gap” sections in the paper, however, the elaboration on what’s the theoretical problems in this area and why your research is urgent is not enough. What have other scholars done on figuring out the role of HRD functions in crisis management? What are the flaws in these studies? What incremental or fundamental contributions the authors could make above and beyond previous wisdom?

Include in literature review some more references related specifically to the crisis management, just 1 or 2 will be enough.

2.Underdeveloped hypotheses development. Arguments in the hypotheses development section are generally thin and weak. For instance, in the reasoning for Hypothesis 1, the authors mainly stated the logic in the first sentence (without any citation). Specifically, the authors were literally saying that training is useful in crisis management because training would “train the workforce on how to manage a crisis.” How exactly training works in crisis management due to Covid-19? For the five stages of your dependent variable, which would be affected most by training practice? And the same with every hypothesis else. The authors need a more in-depth reasoning process to develop the hypotheses.

3.Model configuration. As the authors distinguished between different stages of crisis management, why not test them separately in the analysis and get a derived finding from the basic model? I know that may make the model complex, but the results would be interesting if you can find that different HRD functions work at different stages of crisis management.

Thank you for the suggestion, this will be addressed in the follow up article.

Minor issues:

1.Missing page number in REFERENCES:

Abo-Murad, M., Abdullah, A. K., & Jamil, R. (2019). Effect of the Organisational Culture on Crisis Management in Hotel Industry: A Qualitative Exploration. International Journal of Entrepreneurship, 23(2).

Otoo, F. N. K., & Mishra, M. (2018). Measuring the impact of human resource development (HRD) practices on employee performance in small and medium scale enterprises. European Journal of Training and Development.https://doi.org/10.1108/EJTD-07-2017-0061

Vardarlıer, P. (2016). Strategic approach to human resources management during crisis. Procedia-Social and Behavioral Sciences, 235(2). https://doi.org/10.1016/j.sbspro.2016.11.057

2.Fonts are not unified.

3.Some period (s) were missing. Such as “This is because no firm is immune to the requirement for processes that help to obtain and maintain its capabilities for renewal and stability (Armitage, 2018)”

The page number in references has been included along with the rest of the minor issues.

Reviewer #2: Appreciating your work, please find hereunder some comments:

1. The overall format of the article does not follow PLOS guidelines, for instance line numbers and headings that make it difficult to give feedback. Additionally, it appears that it follows a format for a research proposal than an a research communication manuscript. 1. Please ensure that your manuscript meets PLOS ONE's style requirements, including those for file naming. The PLOS ONE style templates can be found at 

Thank you for your appreciation, we have reviewed the format of the article following the author's guide

N.B. While the English is readable, I would also suggest to have the overall grammatical flow reviewed by a professional in that regard.

The grammar has been reviewed to improve the reading flow. 

2. Abstract lacks a background segment; as it stands, it is a summary of the methods and results. Also, HRD should only be abbreviated after first being mentioned in full capacity.

The abstract has included the background segment and HDR has been mentioned in full capacity.

3. Consider moving third paragraph of the introduction to second to last as the flow correlates more there.

The third paragraph has been moved in order to better correlate with the text flow. 

4. Consider moving last paragraph of the study problem to a separate conclusion segment.

The last paragraph of the study problem has been moved to the newly incorporated conclusion section. 

5. Suggest to summarize aim and objectives, along with those mentioned in the introduction, into one paragraph that ties up the overall flow of the introduction segment.

The aim and objectives of the study have been summarized to improve the overall flow of the introduction. 

6. Suggest to remove literature review as a separate segment and summarize the contents to fit into the introduction segment.

The literature review has been summarized to fin into the introduction segment. 

7. Suggest to broaden the discussion segment to include comparative analysis of other studies, such as those mentioned in the literature review. Matter of fact, it would make the most sense to condense the literature review to narrow down its contents to those that further strengthen the introduction and move a bulk of the literature review contents to the discussion segment for a comparative narration and analysis.

The discussion segment has been broaden, including comparative analysis of other studies and parts of the literature research have been reorganized in the discussion section. 

8. Suggest to have a separate conclusion and discussion segment.

A separate conclusion section has been created. 

9. Can the authors provide their questionnaire in a separate file?

The questionnaire is provided in a separate file. 

10. Can the authors cite the legal framework/IRB protocol that waived consent in this case? I am assuming consent was waived as there is no mention of consent in the manuscript.

The consent form is included within the documentation.

Reviewer 3:

Review Comments to the Author (min 200 characters) see the attachment. ("As the result of the effect of HRD has been proved, it recognizes which dimensions of the crisis management plan should solve and what possible ways to respond. Different levels of values and the uniqueness of jobs may theoretically influence how HRD responds to crisis management (Garavan, 2007). The results prove that Values and Uniqueness do not affect CM mean that upon an increase of those factors, crisis management effectiveness should can be increased. It is the value for the research to explore the impact of certain HRD functions on the effectiveness of crisis management.")

Thank you for your appreciation, the following paragraphs have been included in the conclusions:

Training has a positive and significant influence on crisis management. In other words, with an increase in employee training, there is better crisis management.

Leadership positively and significantly influences crisis management. That is, if the leaders of the organization carry out their task in a way that is respected and the employees follow their guidelines willingly, it leads to better crisis management.

Organizational strategy positively and significantly influences better crisis management. That is, if the organizational management is well defined, it is clearly transmitted to the employees, it generates good crisis management.

The organizational structure does not significantly influence crisis management.

Organizational culture positively and significantly influences crisis management. In other words, a well-defined culture, consolidated among employees and well transmitted, produces better crisis management.

Values and uniqueness of work do not significantly influence crisis management

---

## [Editor Report · Decision Letter 1]

11 Jan 2022

Exploring the role of Human Resource Development functions on Crisis management: the case of Dubai-UAE during Covid-19 crisis

PONE-D-21-34065R1

Dear Dr. Jimber del Río,

We’re pleased to inform you that your manuscript has been judged scientifically suitable for publication and will be formally accepted for publication once it meets all outstanding technical requirements.

Kind regards,

Rogis Baker, Ph.D

Academic Editor

PLOS ONE
---

## [Editor Report · Acceptance letter]

26 Jan 2022

PONE-D-21-34065R1 

Exploring the role of Human Resource Development functions on Crisis management: the case of Dubai-UAE during Covid-19 crisis 

Dear Dr. Jimber del Río:

I'm pleased to inform you that your manuscript has been deemed suitable for publication in PLOS ONE. Congratulations! Your manuscript is now with our production department. 

Kind regards, 

on behalf of

Dr. Rogis Baker 

Academic Editor

PLOS ONE